# Regulatory T Cells Boost Efficacy of Post-Infarction Pluripotent Stem Cell-Derived Cardiovascular Progenitor Cell Transplants

**DOI:** 10.3390/cells14130956

**Published:** 2025-06-23

**Authors:** Aline Derisio de Lima, Hernán Gonzalez-King Garibotti, Qing-Dong Wang, Cecilia Graneli, Tania Incitti, Valérie Bellamy, Maria Eduarda Anastácio Borges Corrêa, Myriam Assal, Makoto Miyara, Jean-Sébastien Silvestre, Karin Jennbacken, Philippe Menasché

**Affiliations:** 1Integrative Physiopathology and Therapeutics of Cardiovascular Diseases, University of Paris Cité, INSERM Unit 970, Paris Cardiovascular Research Center (PARCC), 56 Rue Leblanc, F-75015 Paris, France; derisio.aline@gmail.com (A.D.d.L.); valerie.bellamy@inserm.fr (V.B.); meabcorrea@unesc.net (M.E.A.B.C.); myriam.assal@etu.u-paris.fr (M.A.); jean-sebastien.silvestre@inserm.fr (J.-S.S.); 2Research and Early Development, Cardiovascular, Renal and Metabolism, BioPharmaceuticals R&D, AstraZeneca, Pepparedsleden 1, 43150 Mölndal, Sweden; hernan.gonzalez-king@astrazeneca.com (H.G.-K.G.); qing-dong.wang@astrazeneca.com (Q.-D.W.); cecilia.graneli1@astrazeneca.com (C.G.); tania.incitti@astrazeneca.com (T.I.); karin.jennbacken@astrazeneca.com (K.J.); 3Research and Early Development, Bioscience In Vivo, Respiratory and Immunology, BioPharmaceuticals R&D, AstraZeneca, Pepparedsleden 1, 43150 Mölndal, Sweden; 4Immunology Department, INSERM Unit 1135, Centre d’Immunologie et des Maladies Infectieuses (CIMI), AP-HP, Sorbonne Université, Hôpital Pitié-Salpêtrière, 47-83 Bd de l’Hôpital, F-75013 Paris, France; makoto.miyara@aphp.fr; 5Department of Cardiovascular Surgery, Hôpital Européen Georges Pompidou, 20 Rue Leblanc, F-75015 Paris, France

**Keywords:** immune tolerance, cell therapy, myocardial infarction, regulatory T cells, cardiovascular progenitor cells

## Abstract

Cell therapy is promising for heart failure treatment, with growing interest in cardiovascular progenitor cells (CPCs) from pluripotent stem cells. A major challenge is managing the immune response, due to their allogeneic source. Regulatory T cells (Treg) offer an alternative to pharmacological immunosuppression by inducing immune tolerance. This study assesses whether Treg therapy can mitigate the xeno-immune response, improving cardiac outcomes in a mouse model of human CPC intramyocardial transplantation. CPCs stimulated immune responses in allogeneic and xenogeneic settings, causing proliferation in T cell subsets. Tregs showed immunosuppressive effects on T lymphocyte populations when co-cultured with CPCs. Post infarction, CPCs were transplanted intramyocardially into an immune-competent mouse model 3 weeks after myocardial infarction. Human or murine Tregs were intravenously administered on transplantation day and three days later. Control groups received CPCs without Tregs or saline (PBS). CPCs with Tregs improved LV systolic function in three weeks, linked to reduced myocardial fibrosis and enhanced angiogenesis. This was accompanied by decreased splenocyte NK cell populations and pro-inflammatory cytokine levels in cardiac tissue. Treg therapy with CPC transplantation enhances cardiac functional and structural outcomes in mice. Though it does not directly avert graft rejection, it primarily affects NKG2D+ cytotoxic cells, indicating systemic immune modulation and remote heart repair benefits.

## 1. Introduction

Despite advancements in drug therapies and interventional procedures over the past few decades, heart failure still carries a poor prognosis, which justifies the investigation of new therapies [1]. Cell therapy stands out as disruptive because it not only focuses on alleviating the symptoms, but also addresses the root cause of the disease, which is the loss of millions of functional cardiomyocytes.

While the optimal cell type for myocardial repair remains controversial, cardiac-committed cells differentiated from pluripotent stem cells (PSCs) have progressively emerged as attractive candidates. Although the optimal stage at which these cells should be transplanted remains elusive, cardiovascular progenitors (CPCs), as opposed to more differentiated cardiomyocytes, are credited with a better resistance to stress, a more bioactive secretome, and potential for also generating vascular cells. These cells have now been used clinically without safety issues [2], but still raise the major concern of their allogenicity, currently addressed via pharmacological immunosuppression, whose side effects are well known. These effects can be reduced through a temporary drug regimen when relying on a paracrine mechanism of action that does not necessitate long-term engraftment. However, this approach is not relevant if the goal is the “remuscularization” of the damaged myocardium, which demands the persistence of grafted cells over time. Among the strategies aimed at promoting long-term engraftment via dampening rejection, the induction of immune tolerance via regulatory T cells (Treg) is gaining an increased interest. Treg cells are known for their ability to mitigate immune rejection responses [3] through multiple mechanisms [4], and have shown promising safety and efficacy results in over 60 clinical trials across various pathologies, particularly in solid organ transplantation [5]. The aim of our study has thus been to investigate, in an immune-competent mouse model of post-myocardial infarction (MI) left ventricular (LV) dysfunction, the effects of complementing the intramyocardial transplantation of human CPCs with the systemic delivery of mouse or human Treg cells.

## 2. Methods

### 2.1. Ethics Section

All procedures were approved by the Institutional Ethics Committee of the Université de Paris-Descartes (Project #31295) and by the Animals Ethics Committee at Gothenburg University (Gothenburg Ethical Review Board number EA 1173-2017), and complied with European legislation (European Commission Directive 2010/63/EU) on animal care. Humane care was provided to all animals in accordance with the “Principles of Laboratory Animal Care” formulated by the National Society for Medical Research and the “Guide for the Care and Use of Laboratory Animals” prepared by the Institute of Laboratory Animal Resources and published by the National Institutes of Health (NIH guidelines). All animals undergoing cardiothoracic surgery were anesthetized with isoflurane (2.5 L/min isoflurane at 1.5 L/min of O_2_) and received a pre-operative analgesic, buprenorphine (0.05 mg/kg), administered subcutaneously, which continued for 24 h post surgery. Xylocaine spray was applied to the incision site at the beginning and end of the procedure. Animals used for the isolation of murine Treg cells, echocardiography, and injections received isoflurane (2.5 L/min isoflurane at 1.5 L/min of O_2_). Euthanasia was performed by administering pentobarbital sodium (140 mg/kg). All in vitro tests on human cells were conducted with blood donors who provided informed consent. These tests were performed in accordance with ethical principles originating from the Declaration of Helsinki and were consistent with ICH/Good Clinical Practice, applicable regulatory requirements, and the AstraZeneca policy on Bioethics and Human Biological Samples. The Clinical Study Protocol was approved by the Swedish Ethical Review Authority, including CSP Ed. 3 dated 24 June 2021 (amended from CSP Ed. 2 dated 12 December 2016 with an updated blood sampling process and minor changes from CSP Ed. 1 dated 8 July 2014).

### 2.2. Human Treg Cell Isolation, Expansion, and Characterization

Naïve human (h) Treg cells were isolated and expanded, as previously described [6], with minor modifications. Briefly, blood samples from healthy human volunteers were collected in heparin tubes and incubated with RosetteSep Human CD4+ T Cell Enrichment Cocktail (Stemcell Technologies, Vancouver, BC, Canada), according to manufacturer’s instructions. After incubation, CD4+ cells were extracted via density centrifugation, using Lymphoprep (Stemcell Technologies) and Sepmate tubes (Stemcell Technologies). Enriched CD4+ T cells were incubated with the following conjugated antibodies: CD45RO-FITC (1:3 dilution, clone UCHL1, BD Biosciences, Heidelberg, Germany); CD4-PE-Cy™7 (1:200 dilution, clone SK3, BD Biosciences); CD25-PE/Dazzle™ 594 (1:200 dilution, clone M-A251, BioLegend, San Diego, CA, USA); and CD127-Brilliant Violet 421™ (IL-7Rα) (1:400 dilution, clone A019D5, BioLegend). Naïve Treg cells (CD4+ CD25high CD127low CD45RO-) were sorted on a BD Aria FACS II (BD Biosciences). Sorted naïve Treg cells were seeded at 500,000 cells/cm^2^ in round-bottom 96-well plates in expansion culture media RPMI 1640 Glutamax-1 (Invitrogen, Carlsbad, CA, USA) supplemented with 10% HI-FBS (Invitrogen), 100 U/mL penicillin/streptomycin (Invitrogen), 1 mM sodium pyruvate (Invitrogen), 109 IU/mL recombinant human IL-2 (rhIL-2, Peprotech), and 100 nM rapamycin (Sigma-Aldrich, Burlington MA, USA), and in the presence of Treg cell expander beads (Thermo Fisher Scientific, Waltham, MA, USA) at a bead-to-cell ratio of 4:1. At day 21 of culture, cells were collected, washed, and stained to perform flow cytometry for phenotypic assessment with the following conjugated antibodies: CD39-FITC (1:400 dilution, clone A1, BioLegend); CD25-PE/Dazzle™ 594 (1:200 dilution, clone M-A251, BioLegend); FoxP3-APC (1:32 dilution, clone 236A/E7, Thermo Fisher Scientific); CD152-Brilliant Violet 785™ (CTLA-4) (1:8 dilution, clone BNI3, BioLegend); and Ki-67-BUV395 (1:200 dilution, Clone B56, BD). Samples were acquired using a BD LSRFortessa (BD Biosciences). Afterward, the cells were cryopreserved for subsequent in vivo and in vitro experiments.

### 2.3. Mouse Treg Cell Isolation, Expansion, and Characterization

Mouse (m) Treg cells were isolated from cervical, inguinal, and mesenteric lymph nodes and spleens, collected from immunocompetent C57BL/6J mice aged 8–16 weeks (Charles River, Wilmington, MA, USA). The isolation procedure utilized the mouse spleen dissociation kit (Miltenyi Biotec, Bergisch Gladbach, Germany), followed by the CD4+CD25+ Regulatory T cell Isolation Kit (Miltenyi Biotec), according to the manufacturer’s instructions. The isolation process involved the enzymatic and mechanical homogenization of pooled lymph nodes and spleens, followed by the negative enrichment of CD4+ cells and positive selection of CD25+ cells. Characterization of the isolated m-Treg cells was conducted using a Fortessa X20 flow cytometer (BD Biosciences). The cells were stained with the following antibody panel: CD4-BV421 (1:100 dilution, clone GK1.5, BD Biosciences); CD8-BUV395 (1:100 dilution, clone 53–6.7, BD Biosciences); CD25-PE (1:100 dilution, clone PC61, BD Biosciences); CD73-BV786 (1:100 dilution, clone TY/11.8, BD Bioscience); CD62L-FITC (1:100 dilution, clone MEL-14, Life Technologies, Carlsbad, CA, USA); and FoxP3-APC (1:100 dilution, clone FJK-16s, Life Technologies). To exclude dead cells from the analysis, the Zombie NIR™ Fixable Viability Kit (BioLegend) was employed.

The isolated m-Treg cells underwent expansion using the Dynabeads™ Mouse T-Activator CD3/CD28 for T cell expansion and activation (Thermo Fisher Scientific) in CTS™ OpTmizer™ T cell Expansion SFM media (Thermo Fisher Scientific). The expansion medium was supplemented with 2000 IU/mL of mouse IL-2 IS, premium grade (Miltenyi Biotec), 100 U/mL of penicillin–streptomycin (Thermo Fisher Scientific), 2 mM of L-glutamine (Thermo Fisher Scientific), and 0.1 of mM 2-mercaptoethanol (Thermo Fisher Scientific). Briefly, m-Treg cells were cultured at a density of 500,000 cells/cm^2^ in flat-bottom plates with the Dynabeads™ Mouse T-Activator CD3/CD28 (bead-to-cell ratio of 2:1) for T cell expansion and activation. After 7 days, the cells were cryopreserved for future in vivo and in vitro experiments, with a subset being used to evaluate their immunosuppressive properties and phenotype. The remaining CD4^+^CD25^−^ T cells, obtained as the negative fraction during Treg isolation, were collected and used as responder cells in suppression assays.

### 2.4. Generation and Maintenance of Human Embryonic Stem Cell (ESC)-Derived Cardiac Progenitor Cells (CPCs)

Embryonic stem cells (ESCs) were differentiated into CPCs following the established protocols previously described [7,8], with minor modifications. These modifications included the use of the hESC line SA121 from Takara Bio (Y00025) and the application of 12 µM of CHIR99021 as a GSK3 inhibitor from Tocris. CPC characterization was based on the cardiovascular marker ISL1 (expression of at least ~65%) and Troponin T (cTnT) (expression of at least ~65%). Subsequently, the CPCs were cryopreserved in CryoStor^®^ CS10 from Stem Cell Technologies (Vancouver, BC, Canada).

### 2.5. Suppression Assay

A suppression assay was used to investigate the immunomodulatory properties of Treg cells on human CPCs in both allo- and xenogeneic settings. Mouse effector T cells (Teffs) or human peripheral blood mononuclear cells (PBMCs), labelled with CellTrace Violet (5 µM, CTV, Thermo Fisher Scientific) at 100,000 cells/mL, CPCs were incubated with 50 µg/mL of mitomycin C (Merck, Darmstadt, Germany) for co-culture with both mouse and human cells. for 30 min at 37 °C. First, to assess whether CPCs trigger immune responses, co-cultures were established, with or without 10 ng/mL of IFNγ (PromoCell, Heidelberg, Germany), at 1:1 and 1:3 ratios for mouse Teffs or PBMC and CPCs, respectively. Next, to investigate the effects of Treg cells, co-cultures of CPCs and mouse Teffs were established in the presence of mouse Treg cells, with Teffs–Treg ratios of 1:1, 2:1, 4:1, 8:1, and 16:1. The same setting was repeated in an allogeneic configuration by co-culturing CPCs and human PBMCs from different donors in the presence of h-Treg cells, with PBMC–Treg ratios of 1:1, 2:1, and 1:3. The cells were maintained for 4 days at 37 °C with 5% CO_2_. As controls, both mouse T cells and human PBMCs were cultured in the presence of Tregs alone, without CPC. The proliferation status of CD3+ (CD3-PE, dilution 1:100, clone 17A2, BD Biosciences), CD4+ (CD4-APC, 1:100 dilution, clone A15386, BD Biosciences), and CD8+ (CD8-APC-Cy7, 1:100 dilution, clone RA3-6B2, Thermo Fisher Scientific) cells was evaluated by assessing the CTV dilution through a BD LSR-Fortessa flow cytometer. The data are presented as the percentage of suppression of lymphocyte proliferation. Experiments were performed in technical triplicates.

### 2.6. Mouse Model of MI, Transplantation of CPC, and Treg Cell Injections

First, 12-week-old male immune-competent C57BL/6 mice (Janvier Labs, Le Genest-Saint-Isle, France) were anesthetized with 2.5% isoflurane and the left anterior descending coronary artery was permanently occluded. Three weeks following the induction of MI, mice underwent a baseline echocardiographic assessment of their LV function. Only those with an LVEF ≤ 50% were included. Within one week of this baseline assessment, the selected animals received echo-guided transcutaneous injections of CPC into the peri-infarcted myocardium, with the total dosage distributed across three injection sites, each receiving 10–20 µL. The treatment groups included CPCs (1.4 million cells), and CPCs combined with either human or murine Treg cells, as previously described [9]. The Treg cells were delivered intravenously (via the retro-orbital route) at a dose of 2 million (for both h-Treg and m-Treg cells) in a volume of 150 µL, on the same day as CPC transplantation and 3 days later. Control groups consisted of PBS and Tregs alone (without CPC), also injected intravenously at the same time points. In another set of experiments designed to assess Treg stability post intravenous injection, three mice underwent coronary artery ligation, followed, three weeks later, by the intravenous injection of Thy1.1+ m-Tregs previously expanded in vitro. These m-Tregs were isolated from C57BL/6 mice that express the congenital marker Thy1.1; thus, all the cells from these mice are Thy1.1+ (Janvier Labs, France). Two weeks post injection, the mice were terminated, and Thy1.1+ cells were traced via flow cytometry in the blood, spleen, lymph nodes, and heart. Treg stability was evaluated by measuring the percentage of Thy1.1+ cells expressing Treg markers CD25 and FoxP3.

### 2.7. Echocardiography

Three weeks after MI, mice underwent an echocardiographic assessment of LV function following 2.5% isoflurane sedation using a Visualsonics Vevo 2100 (Visualsonics, Toronto, ON, Canada) imaging system and an MS400 probe suitable for cardiovascular imaging in mice (18–38 MHz). The LV end-diastolic volume (LVEDV), LV end-systolic volume (LVESV), global longitudinal strain of the LV (GLS), and LV ejection fraction (LVEF) were measured from cine loops. The volume data were normalized to mouse body weight (BW) and all measurements were performed on digital loops in triplicates and averaged. Cardiac function was reassessed 3 weeks after CPC or PBS injections. All measurements and subsequent analyses were conducted by an investigator blinded to the treatment group.

### 2.8. Tissue Digestion and Flow Cytometry

Spleens and lymph nodes from mice were processed into single-cell suspensions using 70 µm nylon mesh strainers (Falcon^®^). Erythrocytes in splenocyte samples were removed with a red blood cell lysis buffer (Sigma-Aldrich). Subsequently, single-cell suspensions were prepared for flow cytometric analysis. For NK-cell analysis, spleens were harvested three weeks post CPC injection, while spleens and lymph nodes for murine regulatory T cell tracking were collected two weeks after the transfer of m-Tregs into mice.

Blood samples (50 µL) from mice were collected at the experiment’s conclusion into 5 mL tubes containing 3 mL of PBS (Gibco, Grand Island, NY, USA), with 2 mM of EDTA (Thermo Fisher Scientific), followed by centrifugation at 500× *g* and 4 °C for 5 min. After discarding the supernatant, erythrocytes were lysed by resuspending the sample in a red blood cell lysis buffer (Sigma-Aldrich) and incubating it at room temperature for 10 min. The samples were centrifuged again under the same conditions and underwent a second red blood cell lysis step before being stained for flow cytometric analysis.

Mouse hearts from Thy1.1+ m-Treg-injected mice were exsanguinated via the inferior vena cava, rinsed with sterile PBS (Gibco), and dissected to isolate the left ventricle. The left ventricle was minced, placed in a 1.5 mL tube, and digested for 30–40 min at 37 °C and 1000 RPM agitation using an enzymatic solution of Collagenase II and IV (1 mg/mL each; Gibco), Protease XIV (0.1 mg/mL; Sigma_Aldrich), and DNAse I (16 µg/mL; Roche, Basel, CH) in HBSS with Ca^2+^/Mg^2+^ (Gibco). Digestion was halted by cooling on ice, followed by filtration through 100 µm and 70 µm strainers. Cell suspensions were centrifuged at 400× *g* for 15 min at 4 °C, and the pellet was resuspended in 1–2 mL of FBS-HBSS. Samples were then prepared for flow cytometric analysis.

Flow cytometric staining of tissue samples was carried out at 4 °C for 30 min using the following antibodies: CD45-Alexa Fluor 700 (1:50 dilution, clone 30-F11, BD Biosciences), NKG2D-PE-CFS94 (CD314) (1:100 dilution, clone CX5, BD Biosciences), CD4-APC (1:100 dilution, clone A15386, BD Biosciences), CD4-BV421 (1:100 dilution, clone GK1.5, BD Biosciences), CD3-PE (1:100 dilution, clone 17A2, BD Biosciences), CD25-FITC (1:100 dilution, clone PC61, BD Biosciences), FoxP3-APC (1:100 dilution, clone FJK-16s, Life Technologies), and Thy1.1-BV711 (1:400 dilution, clone OX-7, BioLegend). For the Fc-receptor blockade, an Fc-blocking reagent (BD Biosciences) was used. Intracellular staining was performed using the Intracellular Fixation & Permeabilization Buffer Set (eBioscience, San Diego, CA, USA).

### 2.9. Luminex

Proteins were extracted from non-infarcted LV myocardium using a high-speed homogenizer (T-25 ULTRA-TURRAX, IKA, Staufen, Germany) and cell lysis buffer (Bio-Techne, MN, USA), and the total proteins were determined via Lowry assay (Bio-Rad, CA, USA). Cytokines such as IL-10, IL-6, IL-2, IL-17A, IL-1β, VEGF, and M-CSF (R&D Systems Biotechne, MN, USA) were measured via bead-based multiplex immunoassay and analyzed on Bio-Plex 200 (Bio-Rad). The analyte concentration was calculated using a standard curve (5-parameter logistic (PL) regression) with BioPlex manager software verision 6.2 (Bio-Rad).

### 2.10. Histological Analysis

Three weeks following the transplantation procedures, the mice were euthanized. Explanted hearts were dissected to separate the apex and base, and each tissue block was submerged in Optimal Cutting Temperature (Fisher Scientific, MA, USA) compound, frozen in liquid nitrogen, and stored at −80 °C. Hearts were then cryo-sectioned into 7 µm-thin sections and analyzed for myocardial infarct size (Masson’s trichrome staining) and subendocardial and interstitial fibrosis (Sirius red staining). The subendocardial area was defined as the inner third of the non-infarcted LV area and the interstitial area as the remaining outer two-thirds [10]. The ratios of infarct area to the total LV myocardium area were averaged to calculate the infarct (scar) size. Interstitial and subendocardial fibrosis were calculated as the percentage of red-stained connective tissue relative to the total myocardial area. The number of capillaries were counted and expressed as the number of capillaries per square millimetre (mm^2^) using a staining Griffonia Bandeiraea Simplicifolia Lectin I (20 mg/mL, Vector, CA, USA) conjugated with rhodamine, in conjunction with the nuclear marker 4′,6-diamidino-2-phenylindole (1 mg/mL, DAPI, Calbiochem, France). In order to search for persisting CPCs, hearts from seven CPC-treated mice and one PBS control mouse were entirely sectioned. Cell tracking was performed at both 48 h and 3 weeks post transplantation. Sections from distinct levels of the heart, ranging from th4 apex to base, were analyzed for the presence of human cells using anti-human lamin A/C and anti-human SA121 immunofluorescence staining.

### 2.11. Statistics

All data are expressed as the mean ± standard deviation (SD). Readouts were tested using one-way ANOVAs with GraphPad Prism version 10.1.0.316 (GraphPad Software, La Jolla, CA, USA) and analyzed by an independent statistician blinded to the treatment groups. Multiple comparisons were performed, and post hoc contrasts of the mean change from the baseline between groups were performed using Tukey’s method for pairwise comparisons. *p*-values < 0.05 were considered statistically significant.

## 3. Results

### 3.1. Treg Phenotyping

At the end of the culture period, both m- and h-Tregs expressed the markers CD25 and FoxP3 (Appendix A). Two weeks post injection, Thy1.1+ m-Tregs were detected in the blood, spleen, and lymph nodes of infarcted mice, retaining the Treg markers CD25 and FoxP3 (Appendix A). Thy1.1+ cells were absent in the heart. Notably, the percentage of CD4 and Thy1.1+ m-Tregs in the lymph nodes was higher than in the spleen, and these cells demonstrated the better retention of Treg markers compared to blood and spleen Thy1.1+ m-Tregs.

### 3.2. CPCs Trigger Both Allogeneic and Xenogeneic Immune Responses

When co-cultured with mouse or human lymphocytes, with or without the pro-inflammatory cytokine IFNγ, CPCs induced the proliferation of CD3+CD4+ and CD3+CD8+ T cell subsets. This effect was dose dependent (Figure 1A,B).

Mouse Treg cells exhibited dose-dependent immunosuppressive effects on T cell subsets when co-cultured at various ratios with CPCs and mouse T lymphocytes (Figure 2A). Likewise, h-Treg cells dose-dependently decreased CD3+CD4+ and CD3+CD8+ T cells, stimulated by a 4-day co-exposure of human PBMCs and CPCs (Figure 2B).

### 3.3. Administration of CPCs with Treg Cells Contributes to the Recovery of Cardiac Function

Only the combination of CPC transplantation with h-Treg or m-Treg cells led to a significant improvement in LV systolic function (LVEF, LVESV, and GLS) compared with PBS-injected control mice (Figure 3A–C). However, the LVEDV did not differ between groups (Figure 3D). Mice injected with either m-Tregs or h-Tregs alone (without CPCs) featured functional recovery, which did not differ from that of the control PBS-injected control hearts (Figure 3).

### 3.4. CPCs with Treg Cell Treatment Modulate Splenocyte NKG2D+ Cytotoxic Cell Population

Three weeks after treatments, analysis of the immune cell profile in splenocytes revealed significant reductions in splenic NKG2D+ cytotoxic cells (which include NK cells, activated CD8+ T cells, and γδ T cells) in mice treated with h-Treg or m-Treg cells, compared to those in the PBS group (Figure 4A,B). In contrast, other lymphoid and myeloid cell populations showed no significant differences among the four groups.

### 3.5. The Combination Therapy of CPCs and Treg Cells Alters the Inflammatory Cytokine Milieu in Cardiac Tissue

Mice treated with a combination of CPCs and m-Treg cells exhibited elevated levels of VEGF, along with decreased levels of IL-17A and IL-1β compared to those receiving PBS. Additionally, CPCs with either h-Treg or m-Treg cells exhibited lower concentrations of IL-2 and IL-6 than in the PBS control group. M-CSF concentrations were significantly decreased only in mice treated with CPCs and h-Treg cells compared to the PBS group (Figure 5A,B). There were no notable variations observed among groups regarding the levels of IL-10.

### 3.6. Combining CPCs with Treg Cells Limits Adverse Tissue Remodelling Without Sustained CPC Engraftment

Scar size did not differ between groups (Figure 6A). However, h-Treg or m-Treg cells in combination with CPCs reduced myocardial interstitial fibrosis. Notably, the h-Treg cells group demonstrated less interstitial fibrosis compared to the CPC alone and PBS groups. All treated groups exhibited a decrease in subendocardial fibrosis following 3 weeks of treatment (Figure 6B–E). Angiogenesis significantly increased in all CPC-treated groups, as evidenced by the higher capillary density compared to the PBS group (Figure 6F,G).

While positive controls of human heart tissue and CPC culture showed strong nuclear human lamin A/C and SA121 staining, no human cells were detected in any examined sections of the mouse hearts, suggesting that the grafted cells do not persist up to the 3-week post-treatment time point. Immunostaining was complemented at the 3-week mark by a search for human-specific Alu sequences using real-time reverse transcription polymerase chain reaction (RT-PCR), which failed to detect human-specific gene expression either. We retrieved regulatory T cells from the blood, lymph nodes, and spleen two weeks after intravenous injection in mice post myocardial infarction (Appendix A). However, these cells were not detected in the heart. Additionally, the recovered regulatory T cells retained FoxP3 to varying degrees, with those from the lymph nodes exhibiting the highest retention.

## 4. Discussion

**Key findings.** The main findings of this study are that (1) Treg therapy, both allo- and xenogeneic to the recipient, potentiates the beneficial effects of CPC transplantation, (2) this effect is not related to the prevention of immune rejection, and (3) the benefit seems to be rather related to a modulation of the systemic immune response with a remote positive impact on heart function and structure.

**Rationale for the use of tregs.** Several trials are presently evaluating CPCs or more differentiated cardiomyocytes derived from pluripotent allogenic stem cells, which raises the challenge of preventing their rejection. This is currently achieved via conventional drug-based immunosuppression, which is mandatory to keep the cells alive [11]. However, given the well-known side effects of immunosuppressive drugs, alternate strategies are being eagerly explored. They include the use of haplotypic cell lines which, however, do not avoid an immune response triggered by the persistence of minor alloantigens [12]; gene editing [13,14], still fraught with uncertainties regarding its long-term safety; and the induction of immune tolerance. In this latter context, the use of Treg cells is attractive given their key role in immune modulation and their efficacy track record in auto-immune diseases and solid organ transplantation [15]. However, to the best of our knowledge, the tolerogenic effects of Tregs have not yet been tested in conjunction with CPC transplantation.

**Treg mitigate the immunogenicity of CPCs.** In the present study, we thus tested Tregs from two species to mimic the clinical scenarios in which the recipient’s Treg cells can be used either as donor allo-reactive (following their culture in the presence of the donor’s PBMC, which was grossly simulated in our experiments by the use of human Treg cells, i.e., cells allogeneic to the “donor” CPC) or reinjected after an ex vivo expansion (here simulated by using mouse Treg cells, i.e., cells syngeneic to the recipient). As expected, both mouse and human Treg cells exerted dose-dependent immunomodulatory effects in mixed lymphocyte reaction assays involving mouse and human effector cells, co-cultured with human CPCs.

**Treg fail to enable CPC engraftment.** However, these in vitro data did not translate into the prevention of the rejection of the grafted CPCs since these cells were rapidly undetectable either via immunostaining or PCR. The failure of Tregs to prevent rejection was unexpected, as both mouse and human Treg cells were cultured in the presence of IL-2 at concentrations likely to have shifted their metabolism towards oxidative phosphorylation, rather than glycolysis, which has been linked to increased immunosuppressive effects [16]. A first hypothesis to explain the clearance of the grafted cells is that human CPCs were injected in fully immune-competent mice, and this xenogeneic setting could have mounted a strong immune response, outweighing the tolerogenic effects of the co-delivered Tregs. This hypothesis is supported by the findings that human dopamine neurons transplanted in a Parkinson rat model are rejected despite the concomitant administration of Treg cells, while the cellular grafts could be identified when both neurons and Tregs originated from the same donor and were transplanted in immune-deficient mice. Second, the systemic route for adoptive transfer was possibly suboptimal compared with a direct cardiac co-transplantation with CPCs [17]. Nonetheless, murine Tregs were detected in the blood, spleen, and lymph nodes two weeks after intravenous adoptive cell transfer in infarcted mice. However, the retrieved regulatory T cells displayed some signs of phenotype instability, potentially suggesting a limited efficacy to avoid CPC rejection due to a loss of functionality after adoptive transfer. Finally, MI might have blunted the suppressive function of CD4+ CD25+ regulatory cells, which have been reported to be dysfunctional at three and six months following MI in patients [18]. However, our flow cytometry data documenting the persisting expression of Treg markers (Appendix A) in infarcted mice argue against this hypothesis.

**Tregs potentiate the benefits of CPCs through paracrine signalling.** Even though Tregs failed to prevent the rejection of the grafted CPCs, they still provided incremental functional and structural benefits over CPC transplantation alone, since the combined CPC + Treg groups were the only ones that demonstrated significantly improved outcomes compared with control hearts. The benefit over CPCs alone also manifested as a significant reduction in interstitial fibrosis in the h-Treg + CPC group compared with CPCs alone (Figure 6B), while the reduction in subendocardial fibrosis compared with PBS was of greater magnitude than that seen between CPC alone-treated hearts and controls. Of note, the benefits of adding Tregs to CPCs were similar regardless of whether they were of human or murine origin, although the former tended to perform better. Put together, these data eliminate a CPC-triggered “remuscularization”, and rather point to paracrine signalling, consistent with the current prevailing view of the mechanism of action of transplanted stem cells [8].

**Tregs induce cardioreparative resetting of the immune system.** More specifically, our data suggest a beneficial modulation of the systemic immune response, evidenced by a significant reduction in NKG2D+ cell populations in the spleen. Since NKG2D is expressed on multiple cytotoxic cell types involved in transplant rejection—including NK cells, CD8+ T cells, and γδ T cells—this reduction indicates a broad immunosuppressive effect, which may remotely contribute to the cardiac protection observed in mice receiving Tregs in addition to CPC transplantation. The activation of splenic immune cells has been implicated in the chronic inflammatory response associated with heart failure [19,20] and, conversely, its mitigation might contribute to improved cardiac function. On the same note, the decrease in splenocyte NK cells induced by the systemic administration of mesenchymal stromal cells (MSCs) correlates with such an improvement [21,22]. This decrease in splenic NKG2D+ cytotoxic cells suggests a Treg-induced shift in the endogenous immune cell phenotype, which could have contributed to dampening inflammation [23], an assumption supported by our findings of the reduced levels of four major pro-inflammatory cytokines in Treg-treated mice (IL-6 and IL-2 in both Treg-treated groups and Il-1β and IL-17A in the m-Treg group). Of note, IL-17A is involved in the late remodelling stages after MI by promoting an infiltration of neutrophils and macrophages and an upregulation of pro-inflammatory cytokines and, in comparison with their wild-type littermates, IL-17A-deficient mice demonstrate an improved survival and attenuated LV dilation 28 days post MI [24]. This ability of Tregs to reduce inflammation and subsequent fibrosis aligns with prior studies that have demonstrated that adoptive Treg therapy can mitigate ventricular remodelling [25,26], independently of an “anti-rejection” effect. These anti-inflammatory and matrix-preserving properties of Tregs are likely to only involve myocardial areas with reversible damage, thereby accounting for our finding of an absence of effect on infarct size. This disconnection between altered function and unchanged scar size is not an uncommon observation [27]. Similarly, systemic injections of MSCs decrease splenic NK cells, increase splenic Treg cells, and improve function without reducing infarct size [21].

**Study limitations.** This study has several limitations. First, there is no consensus regarding the optimal timing of Treg delivery, which, in clinical trials [15], ranges from the day of the transplant [2] to 16 months later [28]. In our study, CPC transplantation was delayed until 4 weeks after the index infarction to ensure that most of the acute inflammation had waned, and then before proceeding with a two-stage Treg infusion. We cannot exclude, however, that better outcomes might have been yielded by different timing, involving, in particular, a pre-transplantation Treg delivery to create a tolerogenic environment at the time of CPC implantation. Whether a direct intramyocardial delivery of Tregs, instead of a systemic one, would have resulted in better outcomes, cannot be excluded either. A second limitation is that the mouse data collected in this study are not readily translatable to the clinical setting, since patients with chronic heart failure demonstrate a reduction in Treg cell frequency and function [29,30]. In practice, however, this hurdle could be overcome by including the phenotype and function of Tregs in the screening of patients considered for CPC transplantation, which is an elective procedure. A third limitation is that we do not have direct evidence that the incremental benefits afforded by the combination of Tregs with CPCs were due to the exogenously supplied Tregs and not to an influx of endogenous (mouse) Tregs triggered by the transplanted CPCs. However, several observations argue against this possibility: (1) the recruitment of Tregs can be facilitated by the factors secreted by the transplanted cells, which is the case for MSCs; however, there is no evidence that CPCs have these immunomodulatory properties. In contrast, CPCs appear to be immunogenic (Figure 1); (2) another factor that may enhance Treg recruitment is the cytokine profile of the myocardial environment; particularly, IL-10 can promote Treg accumulation, but, in our study, tissue levels of this cytokine did not differ between PBS- and CPC-injected hearts; (3) if CPC had improved cardiac function only by triggering an inflammatory wound-healing response [31], without any specific benefit related to the injected Tregs, functional data should have been similar in the CPC and CPC + Treg groups, which was not the case; and (4) endogenous Tregs have been shown to only exert modest effects on the post-infarction reparative response [26]. Finally, one could argue that the data might have been confounded by the xenogeneic setting intrinsic to the injection of h-Tregs in mice. The underlying rationale for using h-Tregs was to simulate the situation of a patient receiving his own Tregs following their pre-injection co-culture with the PBMCs of the donor, hence the use of h-Tregs, since the CPCs were of a human origin. It is, however, unlikely that this configuration clouded the results, in that the functional outcomes were similar in the two groups receiving Tregs, regardless of the origin of these cells.

## 5. Conclusions

In conclusion, while the specific set-up of our experiments might have prevented adoptive Treg cell transfer to prevent the rejection of human CPCs in immune-competent mice, this treatment potentiated the cardio-protective effects of CPC transplantation. As the immunoregulatory effects of Treg cells were otherwise clearly confirmed in vitro, it would be worth testing whether Treg cells combined with immunosuppressive drugs can exert an additive anti-rejection effect, thereby allowing their dosing to be reduced and, consequently, the overall safety of allogeneic cell transplantation to be improved (Figure 7).

**Clinical perspective.** Clinical trials are currently testing cardiomyocytes derived from pluripotent stem cells. However, their expected ability to improve heart function is hampered by the requirement for immunosuppressive drugs and their related side effects. To mitigate the immune response triggered by the allogeneic origin of these cells, regulatory T cells (Tregs) are gaining an increased interest in the context of solid organ transplantation. To assess whether their use could be expanded to cell therapy, we tested the effects of Tregs in immune-competent infarcted mice transplanted with cardiovascular progenitor cells, without any additional immunosuppressive treatment. Tregs potentiated the functional and structural benefits of the cardiac cells, but failed to prolong their engraftment in the myocardium; rather, they seemed to act through a modulation of the systemic immune-inflammatory response. Tregs could thus be a safe and effective adjunct to allogeneic cell transplantation by affording reduced dosing of immunosuppressive drugs.

## Figures and Tables

**Figure 1 cells-14-00956-f001:**
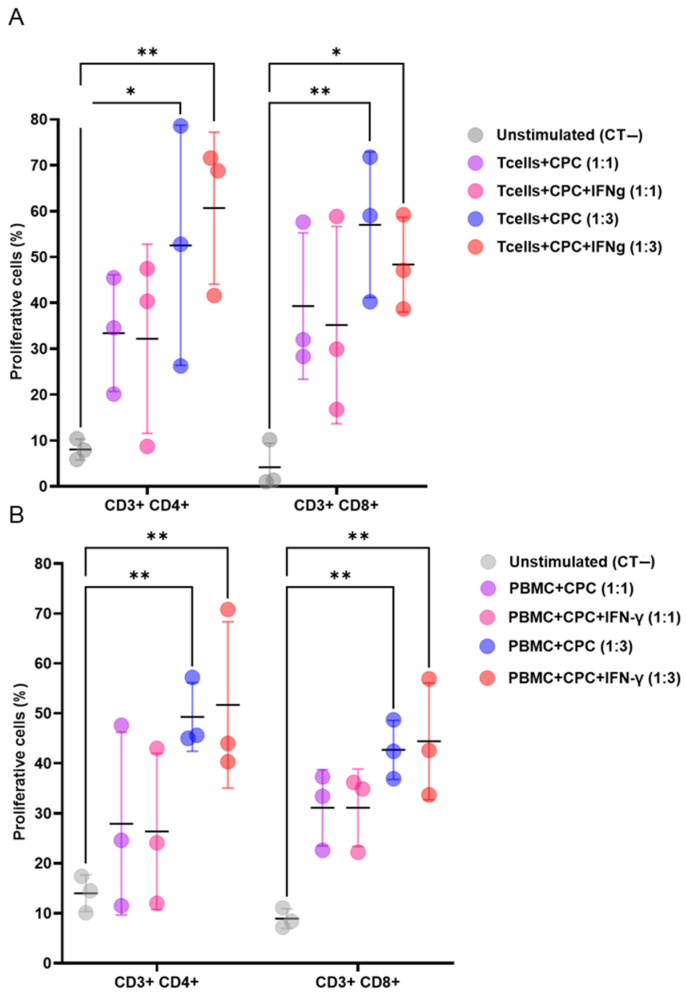
CPCs induce allogeneic and xenogeneic T cell responses. (**A**) CellTrace Violet (CTV)-labelled mouse T cells were cultured alone, or with mitomycin-C-treated CPCs, or IFNγ- and mitomycin-C-treated CPCs at ratios of 1:1 or 1:3. (**B**) CTV-labelled human PBMCs were cultured alone, with mitomycin-C-treated CPCs, or IFNγ- and mitomycin-C-treated CPCs at ratios of 1:1 and 1:3. The levels of CD4+ and CD8+ T cell proliferation were determined by loss in CTV labelling, and are presented as the percentages of proliferating cells. The unstimulated group (negative control, CT−) refers to the co-culture of mouse T cells or human PBMCs with Tregs without the presence of CPCs. Data are presented as mean values ± SD from three independent experiments performed in triplicate. Tukey’s test was used and *p*-values of * <0.05 and ** <0.01 were considered statistically significant. CPCs: cardiovascular progenitor cells; PBMCs: human peripheral blood mononuclear cells.

**Figure 2 cells-14-00956-f002:**
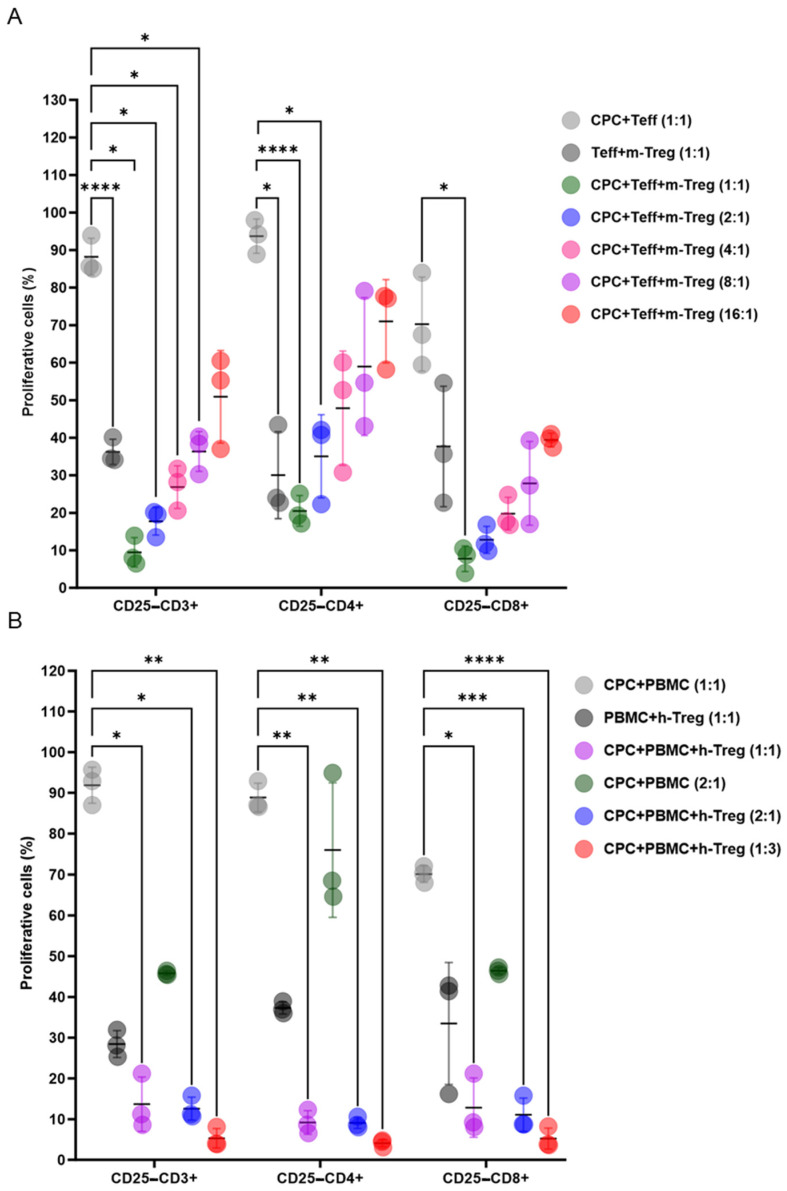
Treg cells exert immunosuppressive effects when co-cultured with CPCs. (**A**) CTV-labelled mouse (m) effector T cells (Teff) and mitomycin-C-treated CPCs were cultured with m-Tregs at ratios of 1:1, 2:1, 4:1, 8:1, and 16:1. (**B**) CTV-labelled human (h) human PBMCs and mitomycin-C-treated CPCs were cultured with h-Tregs at ratios of 1:1, 2:1, and 1:3. The levels of CD4+ and CD8+ T cell proliferation were determined by loss in CTV labelling, and are presented as the percentages of proliferating cells. Data are presented as mean values ± SD from three independent experiments performed in triplicate. Tukey’s test was used and *p*-values of * <0.05, ** <0.01, *** <0.001, and **** <0.0001 were considered statistically significant. CPCs: cardiovascular progenitor cells; Teffs: effector T cells; and PBMCs: human peripheral blood mononuclear cells.

**Figure 3 cells-14-00956-f003:**
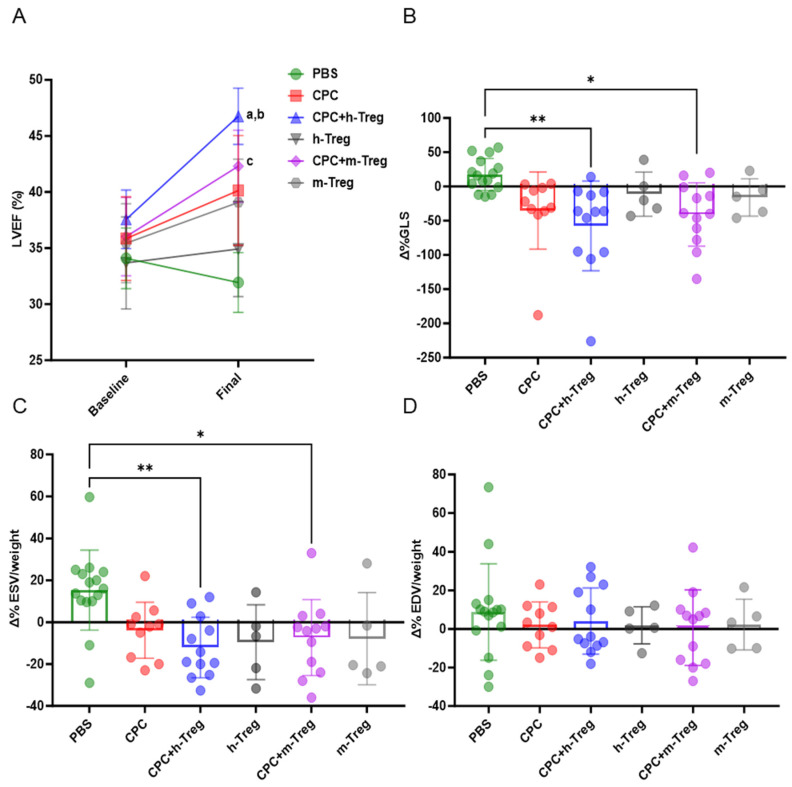
CPCs with Treg cells improve left ventricle function. (**A**) Left ventricle (LV) ejection fraction. (**B**) LV global longitudinal strain (GLS). (**C**) LV indexed end-systolic volumes (ESVs) and (**D**) LV end-diastolic volumes (EDVs) at baseline (3 weeks post MI) and at 3 weeks post CPC injection (7 weeks post MI). PBS (n = 15), CPC-PBS (n = 10), h-Treg (n = 5), CPC-h-Treg (n = 10), CPC-m-Treg (n = 12), and m-Treg (n = 5). GLS, ESV, and EDV data were expressed as delta change from baseline (pre-transplantation, 3 weeks following MI) to the end of study time point. The results are given as mean ± SD. Tukey’s test was used and *p*-values of * <0.05 and ** <0.01 were considered statistically significant. a <0.001 vs. CPC + h-Treg baseline; b <0.05 vs. CPC + m-Treg, and c <0.05 vs. PBS final. BW: body weight.

**Figure 4 cells-14-00956-f004:**
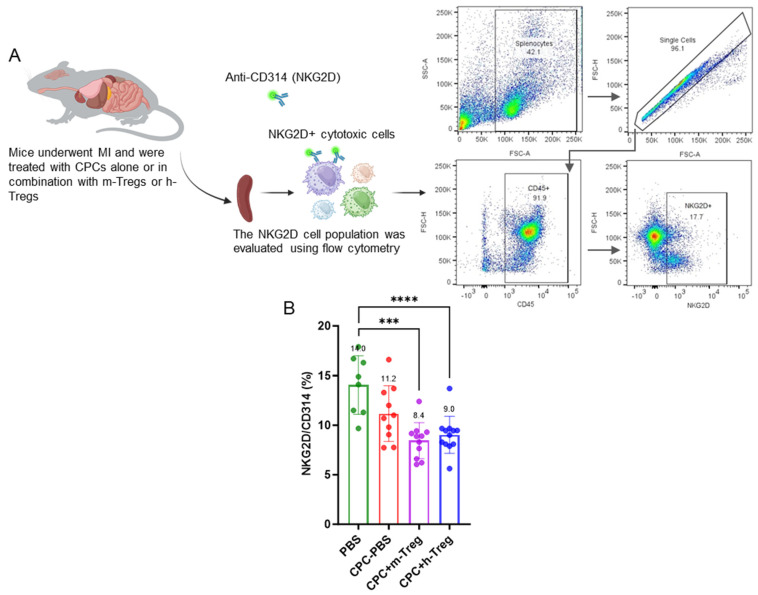
CPC and Treg cell combination reduces splenic NKG2D+ cytotoxic cells. (**A**) Schematic representation of the evaluation of the NKG2D cell population in the spleen collected from mice after myocardial infarction and treatment with either a vehicle (PBS), CPCs alone, or combined with m-Tregs or h-Tregs. Representative flow cytometry plots showing the gating strategy used to quantify splenic NKG2D+, including lymphocyte gating, doublet exclusion, and identification of the NKG2D+ population. (**B**) Flow cytometry assessed the percentage of NKG2D+ cytotoxic cells in the spleen. PBS (n = 8), CPC-PBS (n = 10), CPC-m-Treg (n = 12), and CPC-h-Treg (n = 11). The results are given as mean ± SD. Tukey’s test was used and *p*-values *** <0.001 and **** <0.0001 were considered statistically significant.

**Figure 5 cells-14-00956-f005:**
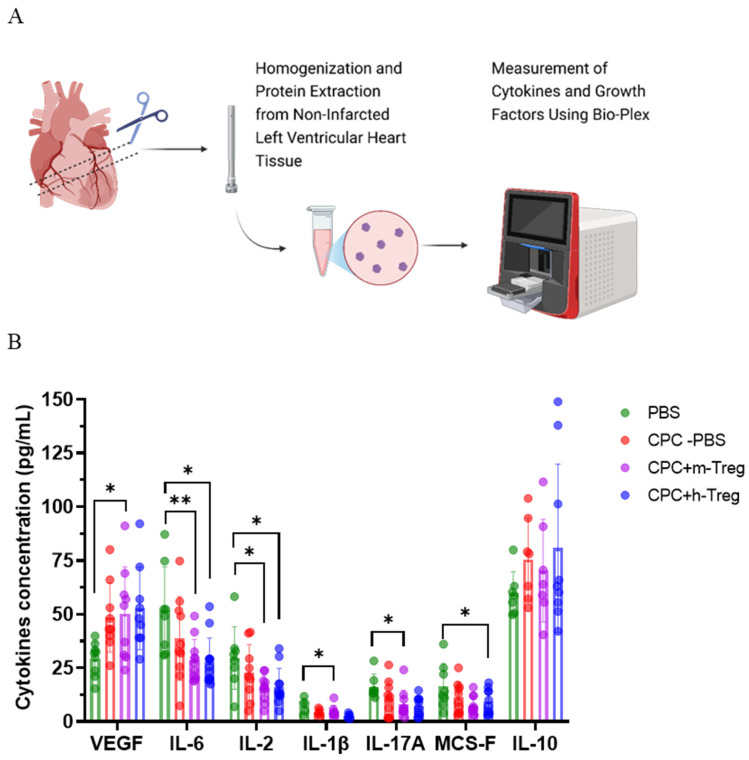
CPC and Treg cell combination changes the heart’s inflammatory environment. (**A**) Schematic representation of the Bio-Plex evaluation of cytokines and growth factors in the heart tissues from the non-infarcted left ventricle of mice after myocardial infarction and treatment with either a vehicle (PBS), CPCs alone, or in combination with m-Tregs or h-Tregs. (**B**) Cytokine concentrations (pg/mL) of VEGF, IL-6, IL-2, IL-1β, IL-17A, MCS-F, and IL-10 in heart tissue using multiplex immunoassay. PBS (n = 8–9), CPC-PBS (n = 9–10), CPC-m-Treg (n = 9–12), and CPC-h-Treg (n = 9–11). The results are given as mean ± SD. Tukey’s test was used and *p*-values * <0.05 and ** <0.01 were considered statistically significant.

**Figure 6 cells-14-00956-f006:**
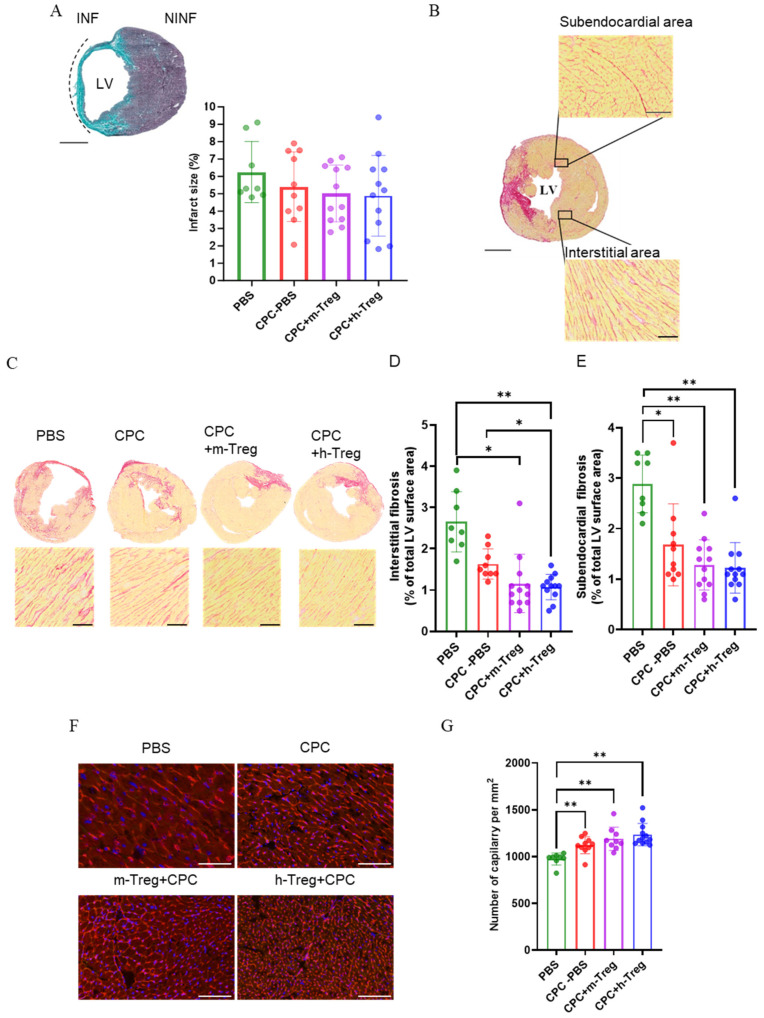
CPC and Treg therapy reduces adverse remodelling in infarcted hearts. (**A**) Masson’s Trichome staining of infarct size. Scale bar = 50 mm. (**B**,**C**) Representative images of Sirius red staining in a transverse section of the heart and collagen volume fraction in the interstitial regions of the LV. Magnification: 20×: scale bar: 250 µm. (**D**) Interstitial and (**E**) subendocardial fibrosis (Sirius red staining of collagen). Scale bar = 50 mm. (**D**,**F**) Representative fluorescence microscopy images of capillaries labelled with lectin–rhodamine red and DAPI-stained cardiomyocyte nuclei (blue). Magnification: 20×. Scale bar: 100 µm. (**G**) Quantification of the angiogenesis as the number of capillaries per mm^2^. PBS (n = 8), CPC-PBS (n = 9–10), CPC-m-Treg (n = 9–12), and CPC-h-Treg (n = 12). The results are given as mean ± SD. Tukey’s test was used and *p*-values * <0.05 and ** <0.01 were considered statistically significant. INF: infarct area; NINF: non-infarcted area.

**Figure 7 cells-14-00956-f007:**
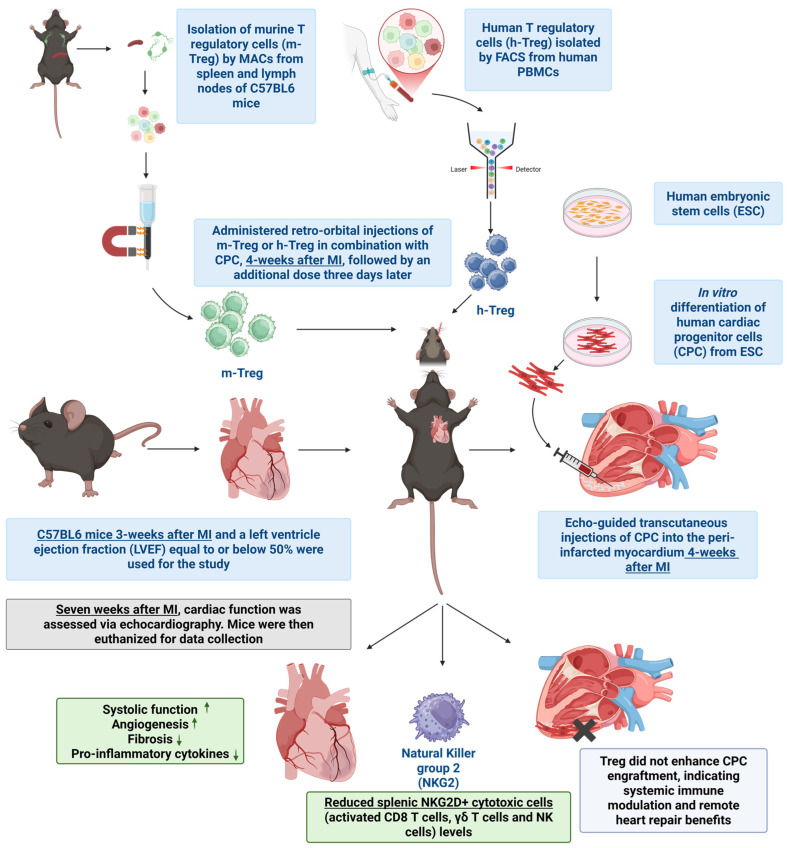
Schematic representation of the proposed mechanism by which CPCs combined with Treg cells improve cardiac function following myocardial infarction, including immune modulation, reduced fibrosis, and enhanced angiogenesis.

## Data Availability

The raw data for each experiment reported can be obtained from the authors upon reasonable request.

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
