# Peer review of "Regulatory T Cells Boost Efficacy of Post-Infarction Pluripotent Stem Cell-Derived Cardiovascular Progenitor Cell Transplants"

_cells, 2025, doi:10.3390/cells14130956_

Round 1
Reviewer 1 Report
Comments and Suggestions for Authors
The manuscript of Lima et al investigates the potency of Tregs to increase eeficacy of CPC reatment after MI. The manuscript is well written and the experiments seem to be technically sound.
My mayor concern is that given the experimental setup it is hard to define efficacy and emchanisms. The author are mixing allogeneic and xenogeneic immune responses with a possible positive effect of Tregs on the regenerative/therapeutical potency of CPC treatment.
As I know that many project evolve over time and of course data (especially when generated with living animals) should be published, the manuscript needs some work to clarify several issues.
The study would very much benefit from syn- and allogenneic controls
Completely syngeneic setup could answer the question if (and at what doses) Tregs (in this setting) improve the regenerative potential of CPCs. An allogeneic murine setup could answer the question whether Tregs could protect allogeneic CPCs in the absence of immunosuppression.
I appreciate the authors stating that the setup might be suboptimal, but the current conclusion drawn from these experiments are not substantiated due to the many variables.
Due to the different species and cells used in the assays the material/method part as well as figures are sometimes confusing/not clear. Starting with the central illustration which does not contain a timeline of the experiment it is hard to understand without having read the whole paper first, A schematic illustration of the experimental setup and timeline would be valuable for the reader (could also be added as supplementary if the central illustration should concentrate also on outcome)
Another important control would be purity after Treg sorting, as stated in material/mjethods it was doen also with more Abs than CD4/CD25/FoxP3 as shown in supp fig1. Also the explanation in the methods section: “with a subset being used to evaluate phenotype..” is not clear whether it refers to before or after cryopreservation (or both).
To me it is very concerning that whereas the authors show over 90% purity for both m and hTregs in suppl fig 1, the purity in suppl fig 2 was not even 80%. Please show purity for ALL experiments and add the number of repeat exp to the figure legends.
In suppl fig 2 it is not clear why the authors did not gate on CD4+ cells for further characterization and why there are CD4 neg cells with positive for the syngeneic marker, please clarify. Protocols for digestion/single cell suspensions from heart, blood and LN are missing
Please state why/how the mitomycin C treatment of CPCs was done. In Fig.1 legend it is not clear whether IFNg treated CPCs have also been treated with mitomycin C (as mitomycin C is never mentioned in the graphs)
Fig 2: please clarify: Teff (graph) correlates to “CTV-labelled mouse (m) T cells? Please explain the abbrevations; protocol for isolation of murine T cells is not in the M&M
Fig.4 please show original data, not a biorender/photostock image for flow plots, a gating strategy would be more useful.
Overall the use of xenogeneic CPCs limits the importance and translational potency of the study as they are immediately rejected and cannot be followed up longterm or substantially influence regeneration of the infarcted area.
Author Response
Comments1: The manuscript of Lima et al investigates the potency of Tregs to increase eeficacy of CPC reatment after MI. The manuscript is well written and the experiments seem to be technically sound.
My mayor concern is that given the experimental setup it is hard to define efficacy and emchanisms. The author are mixing allogeneic and xenogeneic immune responses with a possible positive effect of Tregs on the regenerative/therapeutical potency of CPC treatment.
As I know that many project evolve over time and of course data (especially when generated with living animals) should be published, the manuscript needs some work to clarify several issues.
The study would very much benefit from syn- and allogenneic controls
Completely syngeneic setup could answer the question if (and at what doses) Tregs (in this setting) improve the regenerative potential of CPCs. An allogeneic murine setup could answer the question whether Tregs could protect allogeneic CPCs in the absence of immunosuppression.
I appreciate the authors stating that the setup might be suboptimal, but the current conclusion drawn from these experiments are not substantiated due to the many variables.
Due to the different species and cells used in the assays the material/method part as well as figures are sometimes confusing/not clear. Starting with the central illustration which does not contain a timeline of the experiment it is hard to understand without having read the whole paper first, A schematic illustration of the experimental setup and timeline would be valuable for the reader (could also be added as supplementary if the central illustration should concentrate also on outcome)
Response 1: Thank you for your suggestion. The use of xenogeneic Treg is explained lines 441-447 and further discussed in the last part of the "Limitations" section (lines 521-527).
We have improved the central illustration by incorporating clearer timelines, which we believe will improve readers' comprehension of the experimental outcomes. Below is the original version.
Comments 2: Another important control would be purity after Treg sorting, as stated in material/mjethods it was doen also with more Abs than CD4/CD25/FoxP3 as shown in supp fig1. Also the explanation in the methods section: “with a subset being used to evaluate phenotype..” is not clear whether it refers to before or after cryopreservation (or both).
To me it is very concerning that whereas the authors show over 90% purity for both m and hTregs in suppl fig 1, the purity in suppl fig 2 was not even 80%. Please show purity for ALL experiments and add the number of repeat exp to the figure legends.
Response 2: Thank you for your observation. Supplementary Figure 1A presents a representative plot of CD25 and FoxP3 expression in one batch of human and regulatory T cells used in our in vitro/in vivo experiments. Supplementary Figure 1B summarizes the expression across various batches. The mean expression of CD4, CD25, and FoxP3 in regulatory T cells was approximately 78%. Supplementary Figure 2B shows the expression of CD4, CD25, and FoxP3 in the specific batch used for that experiment, with CD25 and FoxP3 expression around 80% prior to injection, consistent with the mean across batches. To enhance clarity, we have included this information in the Supplementary Figure 1 legend (highlighted in yellow):
“B) The graph displays the mean percentage of cells positive for CD4, CD25, and FoxP3 out of the live cell gate at the end of the in vitro culture period for human and regulatory T cells, represented by blue and purple bars, respectively, across all six independent cell batches used in the in vitro and in vivo experiments.”
Comments 3: In suppl fig 2 it is not clear why the authors did not gate on CD4+ cells for further characterization and why there are CD4 neg cells with positive for the syngeneic marker, please clarify.
Response 3: Thank you for your comments. The experiment results in Supplementary Figure 2 demonstrate that regulatory T cells, expanded in vitro, are detectable two weeks post-adoptive transfer into syngeneic mice. Given that all transferred Tregs express the congenic marker Thy1.1, we concentrated on Thy1.1 and FoxP3 expression. Regarding your comment on CD4 expression loss, this phenomenon is documented in the literature, potentially due to factors like intense cell proliferation or ineffective DNA demethylation in effector cells post-thymic egress1,2.
We realize our results could be better integrated into the main manuscript based on your feedback. Therefore, we added a sentence in the results section “3.6. Combining CPC with Treg Cells Limits Adverse Tissue Remodeling Without Sustained CPC Engraftment” addressing Treg retrieval following adoptive transfer (Starting on line 403): “We retrieved regulatory T cells from the blood, lymph nodes, and spleen two weeks after intravenous injection in mice post-myocardial infarction (Supplementary Figure 2). However, these cells were not detected in the heart. Additionally, the recovered T regulatory cells retained FoxP3 to varying degrees, with those from the lymph nodes exhibiting the highest retention.”; and in the discussion section “TREG FAIL TO ENABLE CPC ENGRAFTMENT,” addressing cell retrieval and phenotype stability in vivo (Starting on line 449): “Nonetheless, murine Tregs were detected in the blood, spleen, and lymph nodes two weeks after intravenous adoptive cell transfer in infarcted mice. However, the retrieved T regulatory cells displayed some signs of phenotype instability, potentially suggesting limited efficacy in preventing CPC rejection due to functional loss after transfer.
Comments 4: Protocols for digestion/single cell suspensions from heart, blood and LN are missing.
Response 4: Thank you for your observation, these protocols have been now included in the materials and methods section “2.8. Tissue Digestion and Flow Cytometry” of the manuscript:
“Spleens and lymph nodes from mice were processed into single cell suspensions using 70 µm nylon mesh strainers (Falcon®). Erythrocytes in splenocyte samples were removed with a red blood cell lysis buffer (Sigma Aldrich). Subsequently, single-cell suspensions were prepared for flow cytometric analysis. For NK-cell analysis, spleens were harvested three weeks post-CPC injection, while spleens and lymph nodes for murine regulatory T cell tracking were collected two weeks after the transfer of mTregs into mice.
Blood samples (50 µL) from mice were collected at the experiment's conclusion into 5 mL tubes containing 3 mL of PBS (Gibco) with 2 mM EDTA (Thermo Fisher Scientific), followed by centrifugation at 500 xg and 4°C for 5 minutes. After discarding the supernatant, erythrocytes were lysed by resuspending the sample in a red blood cell lysis buffer (Sigma Aldrich) and incubating it at room temperature for 10 minutes. The samples were centrifuged again under the same conditions and underwent a second red blood cell lysis step before being stained for flow cytometric analysis.
Mouse hearts from Thy1.1+ mTreg-injected mice were exsanguinated via the inferior vena cava, rinsed with sterile PBS (Gibco), and dissected to isolate the left ventricle. The left ventricle was minced, placed in a 1.5 mL tube, and digested for 30-40 minutes at 37°C and 1000 RPM agitation using an enzymatic solution of Collagenase II and IV (1 mg/ml each; Gibco), Protease XIV (0.1 mg/ml; Sigma Aldrich), and DNAse I (16 µg/ml; Roche) in HBSS with Ca²⁺/Mg²⁺ (Gibco). Digestion was halted by cooling on ice, followed by filtration through 100µm and 70µm strainers. Cell suspensions were centrifuged at 400g for 15 minutes at 4°C, and the pellet was resuspended in 1-2 ml of FBS-HBSS. Samples were then prepared for flow cytometric analysis.
Flow cytometric staining of tissue samples was carried out at 4°C for 30 minutes using the following antibodies: CD45-Alexa Fluor 700 (1:50 dilution, clone 30-F11, BD Biosciences), NKG2D-PE-CFS94 (CD314) (1:100 dilution, clone CX5, BD Biosciences), CD4-APC (1:100 dilution, clone A15386, BD Biosciences), CD4-BV421 (1:100 dilution, clone GK1.5, BD Biosciences), CD3-PE (1:100 dilution, clone 17A2, BD Biosciences), CD25-FITC (1:100 dilution, clone PC61, BD Biosciences), FoxP3-APC (1:100 dilution, clone FJK-16s, Life Technologies), and Thy1.1-BV711 (1:400 dilution, clone OX-7, BioLegend). For Fc-receptor blockade, an Fc-blocking reagent (BD Biosciences) was used. Intracellular staining was performed using the Intracellular Fixation & Permeabilization Buffer Set (eBioscience™).”
Comments 5: Please state why/how the mitomycin C treatment of CPCs was done. In Fig.1 legend it is not clear whether IFNg treated CPCs have also been treated with mitomycin C (as mitomycin C is never mentioned in the graphs)
Fig 2: please clarify: Teff (graph) correlates to “CTV-labelled mouse (m) T cells? Please explain the abbrevations; protocol for isolation of murine T cells is not in the M&M
Response 5: We thank the reviewer for this helpful comment. The aim of the mitomycin C treatment was to inhibit the proliferation of CPCs, so that the readout in the co-culture system would reflect only T cell proliferation. Although CPCs exhibit relatively low proliferation rates, they are still capable of dividing under culture conditions; therefore, mitomycin C treatment was applied to prevent any contribution from CPC proliferation in the co-culture assays. To achieve this, CPCs (including both untreated and IFNγ-treated conditions) were exposed to mitomycin C at a concentration of 50 µg/mL for 30 minutes at 37°C on the first day of co-culture. The cells were then thoroughly washed with PBS to remove residual mitomycin C before being co-cultured with T cells or PBMCs. This approach allowed us to evaluate the immunogenic or immunomodulatory impact of CPCs independently of their own proliferation.
We agree that the information could be clearer in the figure legends. We revised it to clarify that all CPCs used in co-culture experiments, regardless of treatment (±IFNγ), were mitomycin C-treated prior to co-culture. “Figure 1. CPC induces allogeneic and xenogeneic T-cell responses. A) CellTrace Violet (CTV)-labelled mouse T cells were cultured alone, or with mitomycin-C treated CPCs, or IFNγ- and mitomycin-C treated CPC at ratios of 1:1 or 1:3. B) CTV-labelled human PBMC were cultured alone, or with mitomycin-C treated CPC, or IFNγ- and mitomycin C treated CPC at ratios of 1:1 and 1:3.”
We confirm that the term "CTV-labelled mouse (m) T cells" in the figure legend refers to effector T cells (Teff), specifically CD4⁺ and CD8⁺ T cells, used in the proliferation assays. To improve clarity, we have revised the legend accordingly. Revised Figure 2 legend: “Figure 2. Treg cells exert immunosuppressive effects when co-cultured with CPC. A) CTV-labelled mouse (m) CD4⁺ and CD8⁺ effector T cells (Teff)…”
The isolation of murine T cells was performed as part of the Treg isolation protocol, as described in lines 129–156 of the Methods section. Specifically, for suppression assays, conventional T cells (Tconv) were obtained from pooled lymph nodes and spleens of C57BL/6J mice. Following enrichment of CD4⁺ T cells by negative selection, CD25⁻ cells—corresponding to the negative fraction from the Treg isolation process—were collected to obtain a population of CD4⁺CD25⁻ responder T cells. This clarification was added at the end of the corresponding paragraph: ‘The remaining CD4⁺CD25⁻ T cells, obtained as the negative fraction during Treg isolation, were collected and used as responder cells in suppression assays.’”
Comments 6: Fig.4 please show original data, not a biorender/photostock image for flow plots, a gating strategy would be more useful.
Response 6: We appreciate the reviewer’s suggestion. In response, we have replaced the schematic flow cytometry illustration in Fig. 4 with representative original data, now including the complete gating strategy used to identify the NKG2D⁺ cell population. The updated figure shows forward and side scatter plots (FSC/SSC), doublet discrimination (FSC-A vs. FSC-H), and specific gating of NKG2D⁺ (CD314⁺) cells. All data were analyzed using FlowJo software. The figure legend has also been updated accordingly to reflect these changes. We believe this revision enhances the clarity, transparency, and reproducibility of our experimental approach.
Comments 7: Overall the use of xenogeneic CPCs limits the importance and translational potency of the study as they are immediately rejected and cannot be followed up longterm or substantially influence regeneration of the infarcted area.
Response 7: We have justified the use of xenogeneic cells in the revised manuscript (lines 441-447; and lines 521-527).
Reviewer 2 Report
Comments and Suggestions for Authors
The investigators evaluated the effects of combined treatment with cardiac progenitor cells (CPC) and human or mouse regulatory T cells (Tregs) following myocardial infarction. The study addresses the need for novel treatments following myocardial infarction as individuals who survive a myocardial infarction often transition to heart failure later because of continued adverse myocardial remodeling. The investigators demonstrate a significant effect of combined CPC and Treg treatment on cardiac function, particularly left ventricular ejection fraction, and fibrosis. Several suggestions include:
- The investigators illustrate a reduction in NK cells in the spleen following treatment with Treg cells. It would be interesting to evaluate immune/inflammatory cells in the heart, particularly in the non-infarct region where they demonstrate a difference in interstitial fibrosis.
- For functional analyses, (Figure 3), the authors include treatment with Tregs alone. In subsequent figures, this condition is omitted. What was the rationale for not including Tregs alone in these experiments?
Author Response
The investigators evaluated the effects of combined treatment with cardiac progenitor cells (CPC) and human or mouse regulatory T cells (Tregs) following myocardial infarction. The study addresses the need for novel treatments following myocardial infarction as individuals who survive a myocardial infarction often transition to heart failure later because of continued adverse myocardial remodeling. The investigators demonstrate a significant effect of combined CPC and Treg treatment on cardiac function, particularly left ventricular ejection fraction, and fibrosis. Several suggestions include:
- The investigators illustrate a reduction in NK cells in the spleen following treatment with Treg cells. It would be interesting to evaluate immune/inflammatory cells in the heart, particularly in the non-infarct region where they demonstrate a difference in interstitial fibrosis.
- Authors: We thank the reviewer for this insightful comment. Immune cell profiling in the heart was indeed performed in the border zone. However, the analysis yielded a very low frequency of positive immune cell populations in all animals, along with high inter-animal variability. Some immune markers—such as CD3, NKG2D —are considerably more challenging to detect in cardiac tissue compared to lymphoid organs like the spleen. One possible approach to enhance the detection of such populations would be to activate immune cells, for example using lipopolysaccharide (LPS), to boost marker expression. However, this strategy would significantly alter the immune microenvironment and compromise the primary objective of the study, which was to evaluate the effects of CPC treatment, with or without Tregs, under physiologically relevant post-infarction conditions. Due to the high degree of inter-animal heterogeneity, the low expression of certain immune markers, and the limited sample size, the data lacked sufficient statistical power and were therefore not included in the manuscript. We agree that a more detailed analysis of immune and inflammatory cells, would be highly valuable. A larger cohort and extended time points would be necessary to overcome this limitation and yield more robust and interpretable data in future studies.
- For functional analyses, (Figure 3), the authors include treatment with Tregs alone. In subsequent figures, this condition is omitted. What was the rationale for not including Tregs alone in these experiments?
- In our study, we included a group treated with Tregs alone in the functional analyses to assess whether Tregs by themselves could provide a significant improvement in cardiac function following myocardial infarction. This group was primarily included to evaluate the standalone therapeutic potential of Tregs using echocardiographic assessment, as this is one of the most clinically relevant readouts. However, in subsequent analyses focused on histological and immunological outcomes, we chose to prioritize comparisons between untreated MI, CPC alone, and CPC combined with Tregs, as these were central to our study's objective—evaluating the synergistic or additive effects of the combination therapy. Given constraints in animal numbers due to ethical considerations and the limited availability of precious cell products, we prioritized statistical power and biological relevance in our experimental design.
Moreover, previous work has already established the immunomodulatory role of Tregs in the infarcted myocardium (PMID: 22188713), showing that Tregs are recruited to the injured cardiac tissue and can modulate fibroblast function and the local inflammatory response. Therefore, we focused our efforts on novel insights gained from the combination of CPC and Treg treatments.
- References
- Chong MM, Simpson N, Ciofani M, Chen G, Collins A, Littman DR. Epigenetic propagation of CD4 expression is established by the Cd4 proximal enhancer in helper T cells. Genes Dev. 2010, 24, 659-669.
- Teghanemt A, Pulipati P, Misel-Wuchter K, Nouri RV, Scott-Browne JP, Medvedovic J. CD4 expression in effector T cells depends on DNA demethylation over a developmentally established stimulus-responsive element. Nat Commun. 2022, 13, 1477.
Round 2
Reviewer 1 Report
Comments and Suggestions for Authors
My mayor concern is about the experimental setup is still teh same, however, the authors commented on the problem of xenoreactivity and given an explanation for the setup they chose.
The only thing I like to address: as vivible in the revised version, NK cells have been defined by NKG2D only, which in not a valid marker for murine splenic NK cells as it is also expressed on most cytotoxic immune cells (NK cells but also CD8 T cells, gd T cells, NKT T cells,..). So without proper NK cell antibodies, the claims specific for NK cells should be removed.
Author Response
We thank the reviewer for this observation. Indeed, we acknowledge that NKG2D is not a specific marker for murine splenic NK cells, as it is also expressed on other cytotoxic immune cells such as activated CD8⁺ T cells, γδ T cells and NKT cells.
During the course of the study, we initially tested other NK cell-related markers, including NK1.1, as part of our flow cytometry panel design. However, only NKG2D consistently revealed a significant difference between experimental groups in our in vivo model. This pattern aligned with the immunomodulatory effect we observed following the therapy that we tested.
Given this experimental outcome, and to improve the accuracy of our interpretation, we have replaced all mentions of “NK cells” with “NKG2D⁺ cells” throughout the manuscript. This terminology acknowledges the broader population of cytotoxic cells expressing NKG2D and avoids overinterpreting the data as being specific to NK cells alone. Please, see the attachment.
We agree that this change increases the precision of our findings and aligns the terminology with the limitations of the marker used.

Reviewer 2 Report
Comments and Suggestions for Authors
The authors have adequately addressed the previous concerns and I have no further suggestions.
Author Response
Thank you.